# Effects of Microencapsulated Blends of Organics Acids (OA) and Essential Oils (EO) as a Feed Additive for Broiler Chicken. A Focus on Growth Performance, Gut Morphology and Microbiology

**DOI:** 10.3390/ani10030442

**Published:** 2020-03-06

**Authors:** Alessandro Stamilla, Antonino Messina, Sabrina Sallemi, Lucia Condorelli, Francesco Antoci, Roberto Puleio, Guido Ruggero Loria, Giuseppe Cascone, Massimiliano Lanza

**Affiliations:** 1Dipartimento di Agricoltura, Alimentazione e Ambiente (Di3A), University of Catania, Via Valdisavoia, 5, 95123 Catania, Italy; 2DVM consultant poultry specialists, via Cava Gucciardo Pirato, 12, 97015 Modica, Italy; vetmessina@gmail.com; 3DVM consultant poultry specialists, via Alcide de Gasperi, 106, 97013 Comiso, Italy; medvet80@hotmail.it; 4Istituto Zooprofilattico Sperimentale della Sicilia; Via Gino Marinuzzi, 3, 90129 Palermo, Italy; luciacond1980@gmail.com (L.C.); francesco.antoci@izssicilia.it (F.A.); Roberto.puleio@izssicilia.it (R.P.); guidoruggero.loria@izssicilia.it (G.R.L.); giuseppe.cascone60@gmail.com (G.C.)

**Keywords:** organic acids, essential oils, poultry performance, gut health, antimicrobial activity

## Abstract

**Simple Summary:**

Replacing antibiotics with natural alternative compounds in poultry feeding is being increased in the last few years to challenge the antibiotic resistance problem. Among natural compounds, organic acids and essential oils could be a favorable option. The goal of the trial was testing the dietary supplementation of a blend of organic acids and essential oils in broiler diets in order to evaluate growth performance and gut healthiness. The blend of organic acids and essential oils improved growth performances at the end of the growing period and favorably affected, to a certain extent, gut morphology at different gut districts. Moreover, a selective microbial control against *Clostridium perfringens*, *Enterobacteriaceae*, *Enterococci* and *Mesophilic bacteria* was found. Additionally, in litter, organic acids and essential oils dietary treatment drove to an overall decrease of *Mesophilic* bacteria and *Enterococci* counts. Overall, dietary strategy oriented to a supplementation of a mixture of organic acids and essential oils in broiler diets could offer some favorable perspectives in order to maintain adequate growth performance and gut healthiness either in term of morphology or of microbiology. Nevertheless, improving knowledge on the mechanisms of action of these natural additives together with a potential synergistic action is pivotal to clarify their potential as antibiotic replacers.

**Abstract:**

The goal of the trial was testing the effects of a blend of organic acids and essential oils dietary supplementation on growth performance and gut healthiness in broiler chickens. In total, 420 male Ross 308 chicks (1-day old) were randomly assigned to two dietary treatments: basal (BD) and organic acids and essential oils (OA&EO) diets (three replicates/treatment; 70 broilers/replicate). BD group received commercial diets whereas OA&EO group basal diets + 5 g/kg of microencapsulated organic acids and essential oils. OA&EO treatment improved the average daily gain (*p* < 0.01) and feed conversion ratio at 37–47 days compared to BD treatment. OA&EO treatment improved gut morphology mostly at ileum and duodenum levels in terms of villi height, crypt depth, number of villi, mucosa thickness and villi area at 24 and 34 sampling days. A certain selective action against *Clostridium perfringens* in ileum of OA&EO group was shown at 33 (*p* = 0.053) and 46 days (*p* = 0.09) together with lower median values for *Enterobacteriaceae*, *Enterococci*, *Mesophilic bacteria* and *Clostridium perfringens* at ceca level. Overall, organic acids and essential oils supplementation improved growth performance in the final growth stage and some morphological gut traits and reduced to a certain extent *Clostridium perfringens* count in ileum.

## 1. Introduction

The use of antibiotic as growth promoters (AGP) in the poultry industry was widespread all over the world. Initially they were used at low dosage in feedstuffs to improve growth performance and to prevent illness; in Europe, however, their use has been gradually restricted until their total ban starting from 2006 according to European Parliament and Council Regulation EC No. 1831/2003 and European directive 2004/28/EC of the of 31 March 2004 [1,2]. The growth performance of broiler is strictly connected with the healthiness of the gastrointestinal tract in terms of gut immunity, favorable microflora, efficiency in digestion and absorption of nutrients. The AGP banning have driven researchers to focus on alternative approaches to maintain intestinal health, such as the use of pro/pre-biotics, antimicrobial peptides, feed enzymes etc. [3,4,5]. Among additives, organic acids (OA) have long been used for more than three decades to improve growth performance as alternative to AGP in pigs [6] and in broiler chicken [7], although sometimes with controversial results as antibiotic replacers in particular under microbial challenge [8,9]. Generally, a blend of organic acids appeared to be more effective than a single organic acid [7]. 

Essential oils (EO) are mixtures of volatile compounds produced by living organisms and isolated by physical means only (pressing and distillation) from a whole plant or plant part of known taxonomic origin that are proven to exert a natural antibiotic effect with less toxicity and without residues compared to AGP [10]. 

The poultry industry has focused on these additives because they showed an increased nutrients digestion and absorption [11] and a reduction of gut colonization by pathogens [12], together with potential antioxidant properties and a reinforcement of the animal’s immune status [13]. The real effectiveness of these additives in modulating intestinal microflora is strictly connected to the rapid absorption and metabolism upon entering in the intestine [14]. Microencapsulation is a technological strategy to delay the rapid degradation of drugs in the upper gastrointestinal tract [15]. There is evidence that dietary supplementation with microencapsulated blend of OA and phytochemicals additive improved performance in weanling pigs [16]. Moreover, OA mixed with EO improved feed efficiency ratio, intestinal morphology and digestive enzymes activity in broiler chickens [17]. 

Microencapsulated OA and EO, alone or mixed, as feed additive in broiler chickens improved performances and gut microflora [18], reduced intestinal and fecal pathogenic microbial counts [19], lowered the pH in stomach [20], destroyed the cell membrane of pathogens or inhibited their growth [21], improving the activity of digestive enzymes, pancreatic secretion and changed the gut morphology in terms of villus height and crypt depth in small intestine [22]. It is clear that the supplementation of diet with OA together with EO improved the performance of pigs and broiler chickens, as reported in many study [17,23,24]. Then, combinations of organic acids with essential oils can potentially provide a possible synergistic effect in antimicrobial action [25]. Nevertheless, there is a need to improve the number of studies about the effects of antibiotic replacement with OA and EO to modulate the intestinal health in broiler chickens with a focus on gut morphology modifications. The experimental hypothesis was testing if organic acids and essential oils supplementation would improve chicken growth performances and favorably change gut structure and microbial counts due to its antimicrobial role. The goal of this study was then to evaluate a blend of OA and EO microencapsulated as a feed additive on growth performance, epithelial restitution and intestinal microflora in broiler chickens.

## 2. Materials and Methods 

### 2.1. Experimental Design, Animals and Diets

A total of 420 male Ross 308 chicks (1-day-old) from the same hatching were placed in a large shed and were randomly allotted into two treatments with three replicates per treatment (six boxes, 3.5 × 1.5 m) and 70 chicks per replicate. The experimental trial was performed between May and July 2019 according to UE Regulation on using animals for scientific purpose [26]. The dietary treatments were BD (basal diet) and OA&EO (basal diet + 5 g/kg of encapsulated organic acids and natural identical essential oils). The additive used was recognized as improving growth and feed conversion ratio and approved by European Union [27]. It contains organic acids (OA), such as citric (25%, as fed) and sorbic acids (16.7%, as fed), and synthetic essential oils (EO), such as thymol (1.7%, as fed) and vanillin (1%, as fed), and is protected by a matrix coating with a lipid base [28]. Chicks were reared on a comminuted straw-litter and water and feed were automatically supplied ad libitum throughout the experiment by nipples drinkers and plastic feeders. Chicks were vaccinated against Infection Bursal Disease Virus (IBVD) and Infectious Bronchitis (IB) (793b, H120), Marek’s diseases in the hatchery. A coccidiostat (nicarbazin, 40 ppm, and Narasin, 50 ppm) was added to the feeds. No antibiotics were added to diets and water during the trial. They were fed according to a four-phase feeding program with a changed composition through different phases: starter (0–12 d), grower 1 (12–26 d), grower 2 (26–35 d) and finisher (35–47 d), as reported in the Table 1. The composition of nutrients of each basal diet was planned to satisfy nutritional requirements of chicks according to National Research Council [29]. 

### 2.2. Feed Analyses and Chicken Performance

A total of eight BD and eight OA&EO diet samples (two subsamples for each feed for each subperiod subsequently pooled) were collected during the trial, vacuum packed and stored at −30 °C until analyses. Dry matter, crude protein, lipid, crude fiber, ash, calcium, phosphorus, lysine and methionine were analyzed according to Commission Regulation (EC) No 152/2009 [30] that fixed the method of sampling and analyses for the official control of feed. Chicks were weighed at housing (day 0) and at days 12, 25, 35 and 47, in order to calculate average daily gain (ADG) and an average feed intake was recorded for each subperiod to calculate feed conversion ratio (FCR). At every phase change at 11, 25, 34 and 46 days old, three animals per box/replicate (a total of 18 animals per subperiod, nine BD and nine OA&EO) were sacrificed by cervical dislocation according to Council Regulation (EC) No 1099/2009 [31] to sample intestinal content and different gut tracts (duodenum, jejunum and ileum). Moreover, mortality was recorded and reported as an average value along the experimental period. A necropsy in carcasses of dead chicks was also carried out to understand the cause of death.

### 2.3. Intestinal Morphology and Morphometry

Samples for histological analysis of the three segments (duodenum, jejunum and ileum) of the small intestine were immediately taken, after sacrifice, from three chickens for each replicate at every feed change (nine BD and nine OA&EO), for a total of 72 birds. 

Three cm length-portions of each segment were collected during the trial at 11, 25, 34 and 46-day. A total of 216 samples were analyzed for morphometric analysis due to different treatments (BD and OA&EO) and age of chicks. Individual segments of intestine were rinsed with 0.9% of physiological saline and then fixed in 4% buffered paraformaldehyde. The fixed samples were dehydrated, cleared and infiltrated with paraffin in a tissue processor, Leica ASP300, and then embedded in paraffin blocks. The blocks were sectioned in slices of 2.5 µm of thickness, with two slices for every block. The sections were placed on glass slide and stained with hematoxylin and eosin (HE) with an automatic Stainer (Thermo Scientific Shandon varistain gemini) for histological analysis. Morphometric analysis was performed through a Leica DMLB microscope connected with a Nikon camera. Villus height, villus width, crypt depth and crypt width were individually measured on at least five intact villi per intestinal segment. In addition, intestinal mucosa thickness was measured, and villus surface area was calculated using the equation of Rubio et al. [32]: Villus surface area [μm^2^] = π × Villus height [μm] × Villus width [μm] and expressed in Log_10_ basis. The criterion for villus selection was based on the presence of intact lamina propria. Villus height was measured from the tip of the villus to the villus-crypt junction, whereas crypt depth was defined as the depth of the invagination between adjacent villi.

### 2.4. Microbiological Measurements

As for the morphometric measurements, the microbiological analyses were assessed at every feed change (11, 25, 34 and 46 days). Ileal contents of the same 72 chickens used for histological measurements were collected approximately from 1 cm below Merckel diverticulum to 4 cm above caecum tonsils according to procedure of Gheisari et al. [33]; six samples of caecum content were also collected, pooled and homogenized in two samples according to the treatment for each period. Moreover, litter samples were collected along a path of each box marked by a cross, at days 0 (only one sample), 20 and 41 (three samples for each treatment) of the experimental period. At each sampling time, 10 g samples, aseptically weighted, were transferred into a stomacher bag and homogenized with sterile saline water 0.1% (wt/vol) for 2 minutes. Ten-fold dilutions were made and plated in duplicate on the following agar media and conditions: Plate Count Agar (PCA) aerobically incubated at 32 ± 2 °C for 72 h, for all *Mesophilic bacteria*; Violet Red Bile Glucose Agar, aerobically incubated at 37 °C for 24 h, for the *Enterobacteriaceae* count; Tryptone Bile X-gluc agar, aerobically incubated at 40 °C for 24 h, for *Escherichia coli* determination; Bile Aesculine Azide agar, incubated at 37 °C for 24 h, for *Enterococcus* spp. determination; De Man Rogosa and Sharp agar, incubated at 37 °C for 72 h, for *Lactic Acid Bacteria* (LAB) determination; M17 agar, incubated at 37 °C for 72 h, for *Lactococcus* spp. determination; SC agar, incubated in anaerobiosis for 48 h, for determination of *Clostridium perfringens.* Results were expressed as Log_10_ Colony-Forming Unit CFU/g. The detection of *Campylobacter spp*. was assessed through the following protocol: 10 gram samples were enriched in 90 mL of Bolton’s broth and 5% (v/v) of lysed horse blood and subsequently homogenized through stomacher and incubated at 40 °C for 24 h in microaerobic condition. After incubation, 100 µl of the enriched broth was transferred to Modified Charcoal Cephoperazone Deoxycholate Agar (MCCDA) and incubated at 42 °C for 48 h in microaerobic condition. Presumptive *Campylobacter* colonies were observed under phase contrast microscopy (Olympus BX51, Olympus America Inc., Center Valley, PA, USA) for spiral morphology and darting motility [34]. The detection of *Listeria Monocytogenes* has been carried out according to the following protocol: 10 g samples were homogenized through stomacher in 90 ml of Half Fraser broth and aerobically incubated for 24 h at 30 °C. After the incubation, 1 ml of enriched broth eluted in 9 ml of Fraser Broth and incubated for 24 h a 30 °C. After the second incubation, 100 µl of enriched broth was transferred to Aloa Agar and incubated for 24 h at 30 °C [35]. Lastly, for *Salmonella* spp. detection, 10 gram samples were homogenized through stomacher in 90 ml of Tryptone Soya Broth and aerobically incubated for 24 h at 37 °C. After incubation, 1 ml of enriched broth was transferred to Modified Semi-Solid Rappaport-Vassiliadis agar and incubated for 48 h at 40 °C. Presumptive *Salmonella* colonies were transferred on Xylose Lysine Desoxychloate agar and Brilliant Green Agar [36].

### 2.5. Statistical Analysis 

Data on broiler performance were analyzed using a one-way analysis of variance (ANOVA) to test the effect of the dietary treatment (OA&EO vs. BD). Data on gut intestinal morphometry were also analyzed using a one-way ANOVA to test the effect of dietary treatment within sampling day and intestinal section, while microbiological analyses in ileum and litter were processed with the same one-way ANOVA to test the effect of dietary treatment within sampling day. Microbial counts at caecum level were analyzed through a box-plot to show the distribution and the level of bacterial species. Differences between means were assessed using the Tukey’s adjustment for multiple comparisons. Significance was declared when *p* < 0.05, while trends were considered for 0.05 < *p* < 0.10. Statistical analyses were performed by the statistical software Minitab, version 16 (Minitab Inc, State College, PA, USA).

## 3. Results

### 3.1. Growth Performance 

The overall mortality was lower in OA&EO group compared to BD one (2.86% vs. 5.24%, respectively). Animal performances are reported in Table 2 and Table 3. 

No significant difference in live weight between treatments was found at day 12, while at 25 d and at 35 d, the OA&EO group showed a lower (*p* < 0.05) live weight compared to BD. On the other hand, the final weight at 47 d was significantly higher (*p* = 0.001) in the OA&EO group than in the BD one. As a consequence, in the 35–47 d growing phase average daily gain was significantly higher in the OA&EO group (104.5 vs. 78.1 g/d, respectively in OA&EO vs. BD broilers), whereas at 0–12 days, 12–25 days and 25–35 days, no significant differences were found in the growth rate between treatments. Feed intake tended to be lower (*p* < 0.10) at 0–12 days in the OA&EO group compared to its counterpart, while no differences were reported in the subsequent growing phases. Feed conversion ratio improved significantly (*p* < 0.01) in the last growing phase (35–47 days) in the OA&EO group compared to BD according to the differences in the average daily gain previously reported.

### 3.2. Intestinal Morphology and Morphometric Analysis

Morphometric parameters of villi (height, width and surface area) and depth and width of crypts in three segments of the small intestine (duodenum, jejunum and ileum) are reported in Table 4. On day 11, dietary treatment did not affect the morphometric parameters. On day 24 in the ileum tract, significant differences were observed, with villi height (*p* = 0.091) and crypt depth (*p* = 0.055) being higher in the OA&EO group compared to the BD one. Moreover, at the same sampling time, mucosa thickness of duodenum was significantly higher (*p* = 0.002) in OA&EO chicks compared to its counterpart. At day 34 in the ileal tract, OA&EO chicks showed an increased villi height (*p* = 0.001), an increased in tendency (*p* = 0.10) villi width, higher mucosa thickness (*p* = 0.057) and villi area, Log_10_ (*p* = 0.005), compared to BD chicks. At duodenum level, at the same sampling time, the number of villi was higher (*p* < 0.05) in the OA&EO treatment than in BD one. At day 46 in jejunum, villi height tended to be higher (*p* = 0.089) in OA&EO chicks compared to BD ones. Moreover, the epithelial structure of chicks’ gut fed with OA&EO and BD was different in terms of cells organization and inflammation as shown in Figure 1 and Figure 2. In the OA&EO group, villi appear slim, finger-shaped, separated from each other and lined by columnar epithelial cells with a nucleus positioned at the basal third of the cell and visible crypts. They also have a dense network of blood capillaries, lymphatic capillary and connective tissue at the base. In the BD group, intestinal villi resulted merged with neighboring villi, with marked monocyte infiltrate in the central area. Crypts are non-distinguishable.

### 3.3. Intestinal and Litter Microflora Population

Bacterial counts of ileal digesta at different times and for each bacterial family and species are reported in Table 5. There were no significant differences between dietary treatments for all the bacteria analyzed; nevertheless, a lower numerical values of total bacterial count of *Enterobacteriaceae* were found in samples from OA&EO treatment at days 34 and 46, from *Enterococci* at day 11, 25 and 34, from *Escherichia coli* at day 46 and from *Mesophilic bacteria* at day 25, 34 and 46, compared to the BD treatment. As above, in the OA&EO samples, the LAB count increased across all sampling days, although not at a statistical level but only as a numerical value. The bacterial count for *Clostridum perfringens* tended to be lower in intestinal content from OA&EO treatment at day 34 (*p* = 0.053) and at day 46 (*p* < 0.10) compared to the BD one.

Bacterial counts of the caecum content at different days, treatment and for each bacterial species are reported in Figure 3. There were different median values for each species, with a lower median value being recorded in the OA&EO samples than in BD for *Enterobacteriaceae*, *Enterococci*, *Mesophilic bacteria* and *Clostridium perfringens*. The distribution of values showed a different trend, namely the presence of *Escherichia coli* stable in the caecum of OA&EO chickens, while floating in the caecum of chickens fed on basal diet. In litter sampled at different days within treatment, bacterial counts for each bacterial species are reported in Table 6. The most relevant findings occurred on the last day of sampling. Indeed, *Mesophilic bacteria* and *Enterococci* counts were significantly lower (*p* < 0.05) in OA&EO litter compared to BD at day 41. *Enterobacteriaceae* count was lower (*p* < 0.10) in OA&EO at day 20, while an opposite trend was shown at day 41 when the latter bacterial count tended to be higher in the same group compared to BD one. With regard to *Clostridum perfringens* count, the litter of OA&EO group tended to have a lower (*p* < 0.10) total count compared to the BD one. At day 20 of sampling, *Escherichia coli* count was higher in the OA&EO litter compared to the BD one, whereas non-significant differences between treatments occurred at day 41. Within each period in the total of samples of gut tracts and in litter of chicks of both groups, the presence of *Listeria monocytogenes*, *Salmonella* spp. and *Campylobacter* spp. was not noticed.

## 4. Discussion

### 4.1. Growth Performance

The goal of our experiment was testing the feasibility of dietary supplementation at 0.5% dosage of a microencapsulated blend of organic acids, such as citric and sorbic, and phytochemicals, such as thymol and vanillin, in terms of performance, gut integrity and microbial assessment at ileum and caecum level and in litter compared to an antibiotic-free basal diet. The main findings were a reduced mortality rate and an improved growth gain and feed conversion ratio in the last growing subperiod (35–47 days). The supplementation of organic acids and phytochemicals has been well-documented in swine either alone or in combination with or in partial replacement of allopathic growth promoters (antibiotics) due to their acidifying and antimicrobial properties [6,16].

In poultry, few studies focused on the use of a blend of OA and EO and often with controversial results if compared to medicated dietary treatment [28,37,38]. Microencapsulation of a blend of both additives could help to slow the release of these compounds at gut level, thus improving their antibacterial action and, as a consequence, growth performance [38].

Growth performance were favorably affected by OA&EO treatment only at the final growing phase (35–47 days), showing a significant growth recover if compared to previous phases when growth rate decreased in chicks fed an OA&EO diet compared to those fed a basal diet. Probably, a delay in the adaptation to the supplement by OA&EO group that improved later growth performance could justify this finding. Gheisar et al. [28], using different level of supplementation of the same additives, found a linearly increase in body weight gain from day 0 to day 21, while no effect was shown from day 21 up to 35 days as a result of the increase of level of supplementation. Overall, in our trial the dietary supplement at the level of 0.5% (on as-fed basis) did not substantially change the average growth performance, thus confirming that doses, dietary interaction with other ingredients, could have a clear effect on animal performance. It is also possible that overall good environmental conditions with well-nourished chickens and good disinfection practices did not drive to a clear growth improvement in chickens treated with an OA&EO supplement compared to BD ones during the growing period. 

### 4.2. Intestinal Morphology and Morphometric Analysis

The gut ecosystem is composed by three crucial elements: microbial community, intestinal epithelial cells and immune system. These three elements could be affected, positively or negatively, by the diet, gender, background genotype, housing environment, litter and age of birds [39]. Prebiotics such as organic acids and essential oils may not only benefit the intestinal microbiome but also improve the integrity of intestinal epithelial cells, which further increase the absorption of nutrients and enhance the growth performance of animals [40]. In our trial, the supplementation of a microencapsulated organic acids (citric and sorbic acids) and essential oils (thymol and vanillin) improved the morphological parameters after 25 days of feeding with small changes in terms of ileal villi height and crypt depth that tended to be higher and deeper, respectively, in chicken fed with OA&EO compared to BD chicks. Even the muscular tissue of duodenal mucosa started to be much thicker in OA&EO chicks than in BD ones. At day 34, favorable morphological changes of the gut from OA&EO chickens compared to BD birds were clearer, with a higher number of villi in the duodenum segment and higher villi height and villi width in the ileal segment together with thicker mucosa and greater villi area. The increase in crypt depth allows rapid renewal of the villi [41,42]. These findings might explain the improved growth performance in 35–47 day growing subperiod, previously remarked. Due to these findings, the improvement of gut morphology in OA&EO group confirmed that prebiotics are able to enhance the nutrient absorption, thus preserving and improving the intestinal microstructure [43]. It is also known that the increase of villi height and villi width positively affects the area of nutrient absorption [44], thus improving growth performance. 

Bogucka et al. [45] reported positive changes in gut morphology of chickens supplemented with probiotics and symbiotic in terms of higher villus width and greater villus surface area in jejunum and ileum segments and in deeper crypts, thus confirming that bioactive compounds could benefit intestinal morphometric characteristics. Grilli et al. [46] reported a more rapid maturation of the intestinal mucosa by decreasing the local and systemic inflammatory pressure, ultimately resulting in a less permeable intestine in piglets supplemented with the same blend of microencapsulated organic acid and natural identical bioactive compounds such as thymol and vanillin. 

The combining increase of number of villi in duodenum segment together with the improved morphology of ileum tract could infer a more tonic intestine in OA&EO chickens compared to BD birds. Intestinal morphology of epithelial structure in the BD group showed villi fusion, loss of epithelium and presence of monocyte infiltrate; while in the OA&EO gut, structure and morphology appeared regular (Figure 1 and Figure 2).

### 4.3. Modification of Ileal, Caecum and Litter Microflora

The microbiological scenario in terms of the bacterial count of ileum digesta in chicken supplemented with OA&EO did not show significant higher values for most of the bacterial species analyzed compared to broilers fed a basal diet. The *Enterobacteriaceae* count gradually increased in both groups, although any pathogenic strains were observed. The *Enterococci* count, although not different at a statistical level between treatments at each day of sampling, showed less fluctuation across sampling time in chickens supplemented with OA&EO compared to those fed on basal diet. The presence of LAB in the ileal digesta gradually increased in both group without statistical difference, although in chickens fed an OA&EO diet, it was numerically higher compared to BD chickens. Gheisar et al. [28] reported an increased *Lactobacillus* counts in fecal microbiota of chicken fed with the same blend of OA and EO compared to control group. In contrast, the utilization of OA causing several reductions in ileal pH was shown to develop an unfavorable gut environment for the normal proliferation of *Lactobacilli* bacteria [33]. 

The presence of *Escherichia coli* was not affected by dietary treatment, with an exception at day 25, when there was higher amount of *E. coli* in OA&EO chickens compared to BD ones. It is well known that the antimicrobial activity of essential oils such as thymol, eugenol and carvacrol, which showed high antimicrobial activity against pathogenic bacteria such as *Escherichia coli* and *Salmonella typhimurium*, is a potential risk factor of enteric infections [10]. Nevertheless, a blend of organic acids and essential oils may positively affect the presence of *Lactobacillus* but not always cause a reduction of *E. coli* in fecal matter [28]. *Mesophilic bacteria* count showed the same increasing trend across the sampling time with no differences between treatments. 

*Clostridium perfringens* is claimed as the main causative agent of necrotic enteritis (NE) with serious damages in the poultry productive chain [19]. OA&EO chicks showed a dramatic reduction in *Clostridium perfringens* count after 25 days of growing period compared to BD chicks. We could hypothesize a gradual effect of OA&EO supplementation in controlling this pathogen, while in the BD group, a remarkable increase in *Clostridium* counts at 33 and 46 days of sampling was found. The lower presence of *C. perfringens* in ileum of OA&EO broilers is probably connected to the ultrastructural changes promoted by the prebiotic mix, which could increase the resistance capacity of broilers to intestinal infection caused by this pathogen [47]. Furthermore, specific components of EO can inhibit *in vitro* the growth of a number of bacteria, including several strains of *Clostridia* such as *C. perfringens* [48,49]. Mitsch et al., [19] reported *in vitro* an antibacterial effect of essential oils and favorable effects in stimulation of digestive enzymes, and in stabilizing gut microflora due to inactivation of the *C. perfringens* toxins thus reducing its colonization in the broiler gut.

Organic acids are involved in antimicrobial activity through the reduction of digesta pH till in the upper parts of the intestinal tract [49], and it was also shown that the lowering of pH leads to the production of acetic and butyric acids, thus modifying the microflora [50]. In the caeca, all the intestinal refuses accumulated; this gut section is the site of the fermentative activity of most of the bacteria and at the same time the place in which many pathogens establish themselves. The prevalence of anaerobic bacteria, such as *Clostridia*, is well known. Organic acids in the caeca as well as in the small intestine can affect the bacterial counts, as reported in literature [51]. In our experimental trial, the boxplot of bacterial counts in the cecal content showed a peculiar frame. Generally, there were lower median values for *Enterobacteriaceae*, *Enterococci*, *Mesophilic bacteria* and *Clostridium perfringens* in OA&EO caeca compared to BD ones. The range of values was extremely different between treatments if focused on *Escherichia coli*, which, even if it showed a median slightly higher in OA&EO treatment compared to BD, denoted a similar count value along sampling days. *Clostridium perfringens* had a similar range in both the treatment, but showed a lower value in OA&EO chickens.

Organic acids, through lowering pH, protect chickens from various infections, especially at young ages. In addition, they have shown their ability in reducing *Salmonella* colonization in chicken caecum by enhancing the innate immune defense via increased synthesis of host defense peptides [52]. Furthermore, OA also reduced the contamination of litter with harmful microorganisms, neutralizing ammonia production and diminishing the risk of re-infection [53]. We could hypothesize a selective effect of OA&EO treatment against *Mesophilic bacteria* and *Enterococci* counts recorded lower in OA&EO litter compared to BD one, at the last day of sampling. Nevertheless, *Enterobacteriaceae* did not follow the same trend, the concentration being lower at day 20 and higher at day 41 compared to the BD litter. *Escherichia coli* showed a similar trend found in ileum, being higher in OA&EO litter after the first half of the growing period, while comparable onwards with BD litter. Moreover, a certain reduction in *C. perfringens* on the litter, namely the ileum and caecum contents, was reported at the end of growing period.

## 5. Conclusions

A 0.5% supplementation of a blend of microencapsulated organic acids (citric and sorbic acids) and essential oils (thymol and vanillin) in broiler chickens reduced the overall mortality rate and positively affected growth rate in the last period of growing cycle, thus also improving the feed conversion ratio. Moreover, a favorable effect on gut morphology was found in different intestinal segments in the last growing phases. In particular, an increased villi height, villi width, mucosa thickness and villus number after 25 days of supplementation was shown, together with an intact organization of epithelial structures till the end of the fattening cycle. With regard to microbiological aspects, the main finding was a certain reduction of *Clostridium perfringens* count at ileum level in the late growing period and in litter and less *Mesophilic bacteria* and *Enterococci* counts in the litter of broilers fed on diet supplemented with organic acids and phytochemicals. Finally, there is a need to improve the understanding of the mechanism of action of these prebiotics, taking into account the dosage of supplement and potential interaction with other dietary ingredients, and also the great farm to farm variation in their main effects due to different cleanliness of the production environment and heterogeneity of gut microbiota. Moreover, a better comprehension of the synergistic effect of combinations of alternatives to antibiotics as growth promoters is still challenging. 

## Figures and Tables

**Figure 1 animals-10-00442-f001:**
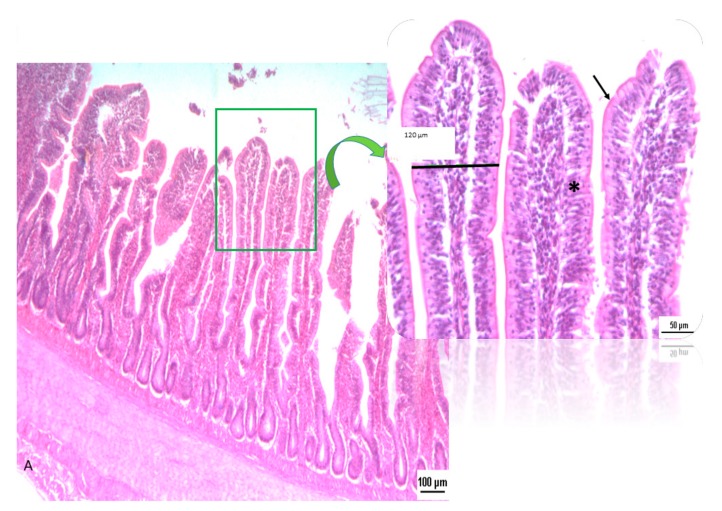
Histological section of OA&EO chicken gut. OA&EO intestinal tract, where villi were slim and finger-shaped. Each villus is lined by columnar epithelial cells with a nucleus positioned at the basal third of the cell (*); at the apex there are microvilli (arrow) that increase the absorbent surface (H&E 5×, 20× magnification).

**Figure 2 animals-10-00442-f002:**
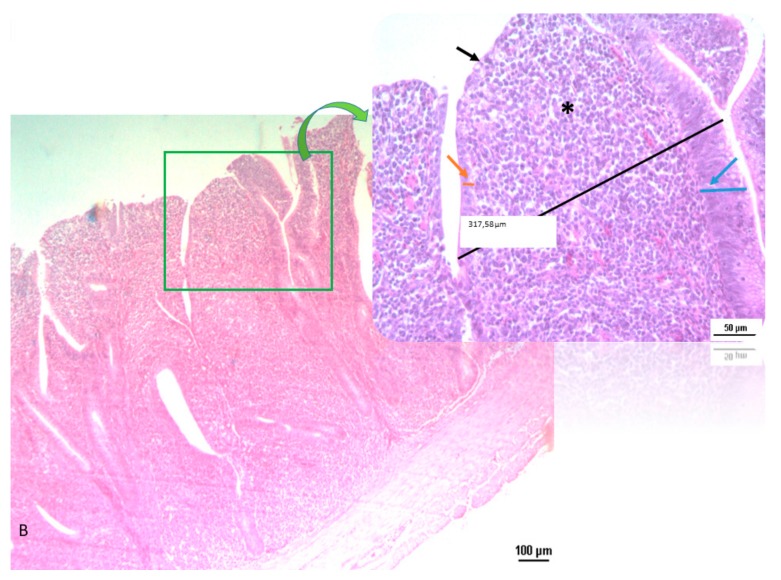
Histological section of BD chicken gut (5×, 20× magnification). In the BD group, villi resulted merged with loss of typical epithelium. In the central area (*), the villus is represented by a monocyte infiltrate; the columnar epithelium is degenerated with some area thinned (orange arrow) and others completely lost (black arrow) or thickened (blue arrow).

**Figure 3 animals-10-00442-f003:**
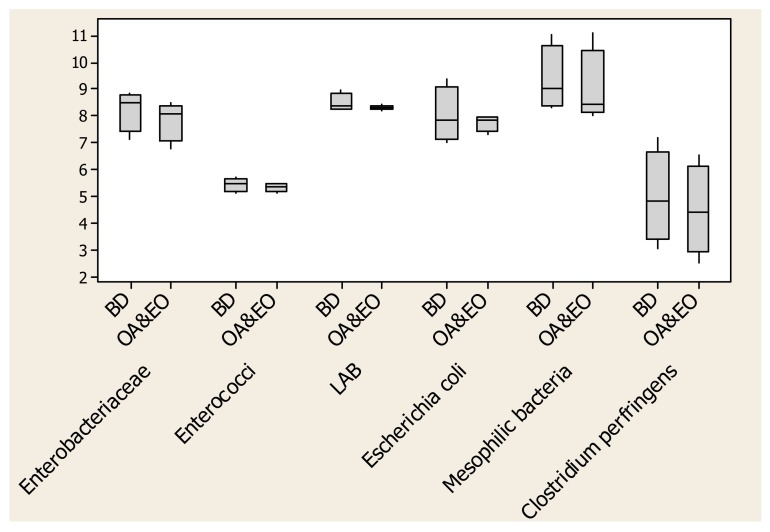
Boxplot of bacterial counts in caeca affected by dietary treatments.

**Table 1 animals-10-00442-t001:** Ingredients and chemical composition of the experimental diet.

Ingredients, g/100 Gas-Fed	Diet			
	Starter (0–12 d)	Grower 1 (12–26d)	Grower 2 (26–35 d)	Finisher (35–47d)
Corn	35	50	51	50
Soybean meal 48%	27.15	28.9	26	23.5
Soybean	10	3	2	2
Wheat	10	0	0	0
Wheat pollard	9	9	10	15
Animal Fat	3.9	4.5	6.4	5.3
Dicalcium Phosphate	1.75	1.5	1.5	1.2
Mineral-vitamin premix ^1^	2.5	2.5	2.5	2.5
Calcium carbonate	0.7	0.6	0.6	0.5
Chemical composition				
Dry matter (DM), g/100g as fed	88.89	88.66	89.26	90.08
Protein, g/100g DM	21.45	19.67	18.76	18.46
Lipid, g/100g DM	8.99	7.23	7.75	7.86
Crude fiber, g/100g DM	3.65	3.07	3.38	3.35
Ash, g/100g DM	5.59	5.61	5.82	5.34
Calcium, g/100g DM	0.87	0.84	0.78	0.63
Sodium, g/100g DM	0.18	0.16	0.17	0.17
Phosphorus, g/100g DM	0.61	0.61	0.59	0.55
Lysine, Lys	1.37	1.44	1.42	1.29
Methionine, Met	0.75	0.76	0.70	0.58
Metabolizable Energy (kcal/kg)	3200	3060	3062	3060

^1^ Provided per kg of premix: copper (9.60 mg), iodium (0.60 mg), iron (60 mg), mangnesium (84 mg), molibdenum (2.4 mg), selenium (0.24 mg), zinc (84 mg), aminoacids (3520 mg), sennic proteasi (15.000 PROT), enzimes (2000 PPU); vitamin A (10.000 UI), vitamin D3 (3.000 UI), biotin (0.12 mg), colin (150 mg), vitamin E (36 mg).

**Table 2 animals-10-00442-t002:** Effects of dietary treatments on growth performance of broiler chicken (n = 90 for treatment).

Treatment	0–12 Days			12–25 Days			25–35 Days			35–47 Days			0–47 Days		
	ADG ^a^	FI ^b^	FCR ^c^	ADG ^a^	FI ^b^	FCR ^c^	ADG ^a^	FI ^b^	FCR ^c^	ADG ^a^	FI ^b^	FCR ^c^	ADG ^a^	FI ^b^	FCR ^c^
BD ^d^	29.79	32.37	1.09	92.39	119.25	1.29	89.82	113.4	1.27	78.151	185.81	2.38	70.66	88.34	1.21
OA&EO ^e^	30.14	31.5	1.05	87.61	117.65	1.34	83.59	111.7	1.34	104.55	195.66	1.88	67.11	86.95	1.24
SEM ^f^	0.318	0.262	0.019	1.61	0.878	0.016	3.05	1.29	0.031	6.45	3.84	0.121	6.77	9.58	0.031
*p*-values	0.635	0.091	0.298	0.152	0.422	0.100	0.363	0.571	0.318	0.01	0.233	0.006	0.802	0.945	0.683

^a^ ADG, average daily gain (g/head/day); ^b^ FI, feed intake(g/head/day); ^c^ FCR, feed conversion ratio; ^d^ BD, basal diet; ^e^ OA&EO, organic acids and phytochemical diet; ^f^ SEM, standard error of means.

**Table 3 animals-10-00442-t003:** Effects of dietary treatments on broiler live weight (n = 90 for treatment).

Treatment ^1^	Live Weight, g			
	12 Days	25 Days	35 Days	47 Days
BD	401.13	1602.18	2500.37	3438.18
OA&EO	405.07	1543.99	2379.88	3634.42
SEM ^2^	3.21	11.8	19.3	29.2
*p*-Values	0.541	0.013	0.002	0.001

^1^ BD, basal diet; OA&EO, organic acids and phytochemicals diet; ^2^ SEM, standard error of means.

**Table 4 animals-10-00442-t004:** Intestinal morphometric analysis as affected by dietary treatment, sampling time and intestinal section (n=9 for each treatment within sampling day).

Items	Day 11				Day 25				Day 34				Day 46			
	Treatment, T ^1^		SEM ^2^	*p*-Value ^3^	Treatment, T ^1^		SEM ^2^	*p*-Value ^3^	Treatment, T ^1^		SEM 2	p-Value ^3^	Treatment, T ^1^		SEM ^2^	*p*-Value ^3^
	BD	OA&EO			BD	OA&EO			BD	OA&EO			BD	OA&EO		
**Villi, n.**																
Duodenum	11.1	10.3	0.889	NS	11.3	9.1	1.1	NS	6.0	9.4	0.871	0.049	5.7	6.2	0.468	NS
Jejunum	14.6	15.4	1.62	NS	13.3	10.8	1.64	NS	10.7	14.0	1.38	NS	8.3	8.6	0.883	NS
Ileum	14.9	16.3	2	NS	17	16.9	2	NS	18.1	18.0	1	NS	10.7	13.6	12.5	NS
**Villi height, VH, μm**																
Duodenum	182,4775	170,7206	6,8104	NS	1,627,299	1,651,397	9,3101	NS	179,1973	194,9865	11,6612	NS	166,2088	181,2208	7,0772	NS
Jejunum	79,1347	76,0604	3,6103	NS	821,632	960,890	6,7449	NS	91,7661	101,1476	6,3316	NS	90,5051	107,8127	5,0823	0.089
Ileum	54,9170	51,9617	1,3956	NS	646,283	767,625	3,5861	0.091	58,6361	76,8206	2,9264	0.001	81,1187	86,2729	6,4917	NS
**Villi width, VW, μm**																
Duodenum	16,6146	16,7485	9500	NS	17,4806	15,3437	9560	NS	19,4824	18,7712	1,8789	NS	22,7073	21,8347	1,5191	NS
Jejunum	14,8624	14,9632	1,0872	NS	13,8523	14,7230	9648	NS	16,8853	18,3951	9788	NS	20,4966	19,2229	1,2640	NS
Ileum	14,0674	12,7699	7,269	NS	14,7288	14,3584	6571	NS	13,6420	178,355	1,2961	0.1	16,4235	20,2750	1,2136	NS
**Crypt, depth CD, μm**																
Duodenum	19,4587	19,0393	1,1623	NS	23,6624	26,7845	1,9466	NS	28,7915	32,8091	1,5754	NS	22,6123	19,7036	2,1050	NS
Jejunum	14,0735	15,7599	1,0492	NS	14,2846	18,3143	1,3830	NS	20,3398	17,8364	1,6135	NS	14,9159	16,8538	1,5840	NS
Ileum	13,2409	12,9022	7489	NS	13,4701	17,6356	1,1009	0.055	13,6884	16,1609	9907	NS	12,2340	12,3774	1,2795	NS
**Crypt, width CW, μm**																
Duodenum	1,5268	1,7226	944	NS	1,8779	1,8635	1014	NS	2,3641	2,4140	1552	NS	1,7381	1,6459	1682	NS
Jejunum	1,7878	1,5713	1144	NS	1,8334	2,0402	918	NS	2,3566	2,2134	1495	NS	1,8076	2,0179	1863	NS
Ileum	1,4568	1,3884	835	NS	2,0121	2,0456	941	NS	2,2037	2,0804	1144	NS	1,8985	1,5580	2373	NS
**IM ^4^ thickness, μm**																
Duodenum	11,8121	11,2260	7393	NS	16,3518	24,8576	1,5416	0.002	15,1654	18,6055	1,1616	NS	11,3496	10,2091	1,2472	NS
Jejunum	10,0921	8,5737	9282	NS	19,5494	15,7568	1,9865	NS	12,5443	15,0728	1,2978	NS	8,5002	7,8229	9276	NS
Ileum	12,3312	11,8648	8753	NS	21,7409	23,8342	1,4278	NS	15,7826	20,6350	1,2914	0.057	13,2033	12,2037	1,1551	NS
**Villi area, Log_10_μm ^2^**																
Duodenum	1.20 × 10	1.19× 10	2.90 × 10^−2^	NS	1.19 × 10	1.19 × 10	3.40 × 10^−2^	NS	1.20 × 10	1.20 × 10	4.60 × 10^−2^	NS	1.21 × 10	1.21E+01	3.10 × 10^−2^	NS
Jejunum	1.15 × 10	1.15 × 10	4.20 × 10^−2^	NS	1.15 × 10	1.16 × 10	4.40 × 10^−2^	NS	1.17 × 10	1.17 × 10	4.50 × 10^−2^	NS	1.17 × 10	1.18E+01	3.80 × 10^−2^	NS
Ileum	1.14× 10	1.13 × 10	2.50 × 10^−2^	NS	1.15 × 10	1.15 × 10	2.80 × 10^−2^	NS	1.14 × 10	1.16 × 10	4.20 × 10^−2^	0.005	1.16 × 10	1.17E+01	5.30 × 10^−2^	NS

^1^ Treatment, T = BD, basal diet; OA&EO, organic acids and phytochemicals diet; ^2^ SEM, Standard error of means; ^3^ p-value: probability level; ^4^ IM, Intestinal mucosa.

**Table 5 animals-10-00442-t005:** Bacterial count of the ileal tract as affected by dietary treatment and sampling time (n = 9 for each treatment within sampling day).

Bacterial Group ^2^	Treatment			SEM ^1^	*p*-Value
	Day	OA&EO	BD		
*Enterobatteriaceae*	11	6.959	6.959	0.145	0.641
(Log_10_ CFU/g)	25	6.707	6.159	0.202	0.202
	34	8.694	8.790	0.139	0.769
	46	8.144	8.415	0.273	0.673
*Enterococci*	11	5.651	6.138	0.260	0.409
(Log_10_ CFU/g)	25	5.924	5.726	0.096	0.360
	34	5.827	6.041	0.230	0.691
	46	5.996	5.612	0.147	0.222
LAB ^3^	11	7.623	7.301	0.180	0.433
(Log_10_ CFU/g)	25	8.559	8.417	0.058	0.259
	34	8.573	8.334	0.110	0.327
	46	9.403	9.085	0.131	0.268
*Escherichia coli*	11	5.398	5.401	0.169	0.992
(Log_10_ CFU/g)	25	7.675	6.729	0.283	0.087
	34	6.526	6.490	0.141	0.914
	46	7.476	7.655	0.173	0.658
*Mesophilic bacteria*	11	7.317	7.145	0.197	0.710
(Log_10_ CFU/g)	25	7.713	7.693	0.426	0.984
	34	8.924	9.196	0.223	0.601
	46	8.306	8.578	0.152	0.432
*Clostridium perfringens*	11	5.908	5.735	0.057	0.140
(Log_10_ CFU/g)	25	1.985	1.667	0.537	0.802
	34	1.360	5.066	1.030	0.053
	46	1.230	4.154	0.879	0.090

^1^ SEM = standard error of means; ^2^ Expressed as CFU = Colony-forming unit; ^3^ LAB = Lactic Acids Bacteria.

**Table 6 animals-10-00442-t006:** Bacterial count of litter as affected by dietary treatment and sampling time (n = 1 for treatment at day 0; n = 3 for treatment at day 20 and at day 41).

Bacterial Group	Treatment			SEM ^1^	*p*-Value
	Day	OA&EO	BD		
*Mesophilic bacteria*	0	5.681	5.681	0	
(Log_10_ CFU/g)	20	9.449	9.501	9.475	0.367
	41	9.392	10.328	0.239	0.022
*Enterococci*	0	3.041	3.041	0	
(Log_10_ CFU/g)	20	8.413	8.317	8.3651	0.502
	41	7.804	8.242	0.108	0.013
*Enterobatteriaceae*	0	3.079	3.079	0	
(Log_10_ CFU/g)	20	8.324	9.067	8.695	0.096
	41	8.338	7.549	0.24	0.096
*Clostridium perfringens*	0	0	0	0	
(Log_10_ CFU/g)	20	0.534	0	0.267	0.374
	41	3.303	4.836	0.460	0.089
*Escherichia coli*	0	0	0	0	
(Log_10_ CFU/g)	20	7.766	6.580	7.173	0.013
	41	7.454	7.031	0.172	0.257

^1^ SEM = standard error of means.

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
