# Peer review of "Effects of Microencapsulated Blends of Organics Acids (OA) and Essential Oils (EO) as a Feed Additive for Broiler Chicken. A Focus on Growth Performance, Gut Morphology and Microbiology"

_animals, 2020, doi:10.3390/ani10030442_

Round 1
Reviewer 1 Report
This manuscript presents data from a single experiment in which broiler chickens were provided basal feed and the same basal feed supplemented with encapsulated organic acid and essential plant oils. The experiment was carefully performed though unlike authors’ opinion, I think that there was no real effect of the tested supplementation. Authors sometimes refer to significant differences but they also had to help themselves also by a term “a trend” indicating differences with p values between 0.05 and 0.1. I do not mind this as long as appropriate discussion is provided. Even negative results are results and may help other authors. Next, I am critical to the quality of English. Some parts are written with laboratory slang and many sentences do not follow basic grammatical structure of English. I strongly recommend that a native speaker experiences in animal science reads and correct this text.
Points for authors consideration
l.25, how many is “several”?
l.44, word “litter” appears as the last one in the abstract although nothing on litter has been mentioned above. Please, correct, say something on litter in the body of the Abstract or delete it also from the conclusive sentence.
l.78, depth relates to crypt and villus, isn’t it? Please correct
l.85, .... AND intestinal microflora...
l.88, delete “In a large shed,”
l.93, add right brackets after “oils”
l.117, “Nonadecanoic acid was used as internal standard” is correct structure of the sentence. This type of grammatical error is present in many other places of this manuscript. Ask a native speaker to correct this throughout.
l.117 and 118, what gas chromatography? Please reword.
l.137, this is not a correct sentence “Leica DMLB connected with camera Nikon; morphometric analysis was performed.”
l.138-139, as above, this is not a correct sentence structure - Villus Height, villus width, crypt depth, crypt width and the thickness of mucosa, at least 5 villi per intestinal segment and only intact villi, from each individual chick were measured.
l.149, English again - then there was used one gram diluted 1:9 (wt/vol) in sterile saline
l.152, First, poor English and second, do not use colons throughout the whole manuscript. Use standard sentences with subject, verb, object etc. “There were evaluated: Mesophilic...”
This comment is valid for the whole manuscript and in particular for this chapter, i.e. 2.4. Microbiological Measurements.
l.157, “... withdrawal 1 ml of solution...” where is subject, where is verb? This is terrible laboratory slang.
l.163, “determination; SC agar incubate in anaerobiosis“, please do not recommend what one should do. Write the text in plain sentences.
l.168, use English, not Italian, „Caseina Cefoperazone Desossicolato Agar MCCDA“
l.172, remove the punctuation after „h“. Hours are usually abbreviated without punctuation
Table 2, the most important are final data, 0-47 days. There are no significant differences between control and experimental chickens.
Table 3, though I will now deny my previous comment, although there is significant difference on day 47, there are also significant difference on days 25 and 35, but in opposite direction in comparison to day 47, i.e. higher body weight in control chickens. When this happens to me, in my experiments, this indicates random fluctuations of no biological meaning
l.203, “although differences in live weight.” Use sentences with subject and verbs
l.220-222, I cannot see any difference in these two figure, please say clearly in the text
l.224, if you want to use this style, definitively all results ARE presented in Table..., not WERE. Correct throughout the whole manuscript.
l.22p, poor English, “... it was recorded an increase...”
Table 4, if I check day 46, all comparisons are insignificant.
l.286, absorption, not adsorption
l.290, “, resulting higher and deepest,” please use standard sentences
295, both references should in a single pair of left and right bracketsDiscussion, authors commonly use generic term “essential oils” but different authors can use different essential oils and their activities may extensively differ, similarly as effect of penicillin and enrofloxacin differs – despite the fact that both are antibiotics.
Reviewer 2 Report
The authors tested natural alternatives to the use of antibiotics on a sample of 308 chickens. In particular, they used a diet enriched with organic acids and essential oils (OA-EO) compared to animals with a basal diet.The use of OA-EO has had the effect of increasing the absorbing surface of the gastrointestinal tract (higher and larger area villi) and modifying the microbiota. In particular, the diet reduces the content of Clostridium perfringens.
Animal mortality drops from 5.24% to 2.86%. Which are, however, very low mortality rates, which would not justify the need to introduce a diet that still has an additional cost compared to a basal diet. (Out of 100 animals on a basal diet five die, with the AE-OA diet three die. Does the death of two chickens cost more or less than the supplement added to the diet of 100 chickens?).
What I find very interesting in this study is the increased absorption surface of the gastrointestinal tract of chickens treated with the OA-EO diet. This phenomenon could play a role in nutrient absorption and therefore have a significant impact on meat quality. This point, which has not been clarified, seems to me to be of great interest.
The microflora also underwent some variations, and this would be interesting to investigate longer in the time. In this sense, it would be also interesting to test other genotypes with a different type of growth.
In conclusion. The study has an adequate number of animals, was conducted in a linear manner and laid the basis for further investigation. If data on the composition of the meat had been added, and if a difference had been found between animals grown on a basal and enriched diet, this article would gain more value.
Minor revisions:
1) The abbreviations in the abstract have not been explicated. Remove abbreviations or make them explicit.
Major revision
1) line 310 "Also looking at Figures 1 and 2 it was noticed the great differences in terms of morphology of epithelial structure between OA&EO and BD intestines, with villi fusion and loss of functional epithelium with presence of inflammatory cells in BD gut while structure, morphology and function appeared regular in OA&EO intestinal segments." Hematoxylin and eosin staining does not identify inflammatory cells that infiltrate the lamina propria or underlying layers. An immunohistochemical assay with a specific antibody is required in order to confirm the presence of immunocytes. For example, if mast cells need to be detected, an antitryptase antibody should be used. In this sense it is better to remove the sentence or add the image of the staining (the immunocyte count should also be added).
Round 2
Reviewer 1 Report
Since the authors accepted most of my recommendations, I agree with its publication in Animals.
Reviewer 2 Report
Authors replied to the comments without actually adding anything to the work.
I think that the article as it is now, without data on the quality of the meat and without adequate recognition of the immunocytes that infiltrate the mucosa, remains of moderate / little interest. In any case the current version can be published.